# Predicting the Area under the Plasma Concentration-Time Curve (AUC) for First Dose Vancomycin Using First-Order Pharmacokinetic Equations

**DOI:** 10.3390/antibiotics12040630

**Published:** 2023-03-23

**Authors:** Kritsaporn Sujjavorakul, Wasan Katip, Stephen J. Kerr, Noppadol Wacharachaisurapol, Thanyawee Puthanakit

**Affiliations:** 1Department of Pediatrics, Faculty of Medicine, Chulalongkorn University, Bangkok 10330, Thailand; k.sujjavorakul@gmail.com; 2Critical Care Excellence Center, King Chulalongkorn Memorial Hospital, Bangkok 10330, Thailand; 3Department of Pharmaceutical Care, Faculty of Pharmacy, Chiang Mai University, Chiang Mai 50200, Thailand; wasankatip@gmail.com; 4Epidemiology Research Group of Infectious Disease (ERGID), Chiang Mai University, Chiang Mai 50200, Thailand; 5Biostatistics Centre, Faculty of Medicine, Chulalongkorn University, Bangkok 10330, Thailand; stephen.k@chula.ac.th; 6The Kirby Institute, The University of New South Wales, Kensington, NSW 2052, Australia; 7HIV-NAT, The Thai Red Cross AIDS Research Centre, Bangkok 10330, Thailand; 8Center of Excellence in Clinical Pharmacokinetics and Pharmacogenomics, Department of Pharmacology, Faculty of Medicine, Chulalongkorn University, Bangkok 10330, Thailand; noppadol.w@chula.ac.th; 9Center of Excellence for Pediatric Infectious Diseases and Vaccines, Faculty of Medicine, Chulalongkorn University, Bangkok 10330, Thailand

**Keywords:** vancomycin, therapeutic drug monitoring, pharmacokinetics, area under the concentration curve, critically ill patients

## Abstract

To treat critically ill patients, early achievement of the target area under the plasma concentration-time curve/minimum inhibitory concentration (AUC/MIC) in the first 24 h is recommended. However, accurately calculating the AUC before steady state is an obstacle to this goal. A first-order pharmacokinetic equation to calculate vancomycin AUC after a first dose of vancomycin has never been studied. We sought to estimate AUC using two first-order pharmacokinetic equations, with different paired concentration time points, and to compare these to the actual first dose vancomycin AUC calculated by the linear-log trapezoid rule as a reference. The equations were validated using two independent intensive first dose vancomycin concentration time data sets, one from 10 adults and another from 14 children with severe infection. The equation with compensation for the alpha distribution phase using a first vancomycin serum concentration from 60 to 90 min and the second concentration from 240 to 300 min after the completed infusion showed good agreement and low bias of calculated AUC, with mean differences <5% and Lin’s correlation coefficient >0.96. Moreover, it gave an excellent correlation with Pearson’s r > 0.96. Estimating the first dose vancomycin AUC calculated using this first-order pharmacokinetic equation is both reliable and reproducible in clinical practice settings.

## 1. Introduction

Vancomycin is a commonly used broad-spectrum antibiotic which covers multidrug-resistant Gram-positive bacteria, e.g., methicillin-resistant staphylococci, penicillin-resistant *Streptococcus pneumoniae* (PRSP), and ampicillin-resistant enterococci [1]. The current guidelines on therapeutic drug monitoring (TDM) for vancomycin recommend achieving the target area under the plasma concentration-time curve (AUC) within the first 24–48 h. This approach maximizes the therapeutic effect, while simultaneously minimizing adverse effects, especially vancomycin-associated acute kidney injury (vAKI) [2]. Early TDM and early dose adjustment are especially important in critically ill patients whose mortality rates are high during the initial phase of illness. For example, between one-third and half of deaths in patients with severe sepsis and septic shock occur within the first 48 h after diagnosis [3,4,5]. However, in clinical practice, routine TDM for vancomycin is measured at steady state at least 24 h after the initial dose. This limits the opportunity to optimize the AUC during critical periods of illness.

The largest pooled population-pharmacokinetic modeling study characterized vancomycin pharmacokinetics as a two-compartment distribution model [6]. There is no apparent metabolism [7], and the absolute elimination rate is a linear function of its concentration in plasma following first-order kinetics [8].

Three methods are commonly used to estimate the AUC: the linear-trapezoid rule, Bayesian-derived AUC monitoring, and first-order pharmacokinetic equations. An accurate collection of multiple concentrations over the same dosing interval is required for the linear-trapezoid rule, making this method unsuitable outside the research setting. Bayesian-derived AUC monitoring is recommended in vancomycin TDM. However, a well-developed population-specific pharmacokinetic model of vancomycin as the Bayesian prior is required, and thus Bayesian dose-optimizing software is not available in some special populations, for example obese patients, critically ill patients, pediatric patients, and patients with unstable renal function [2]. The first-order pharmacokinetic equation method makes fewer assumptions than the Bayesian approach and is simple enough for routine clinical practice use. The advantage of this method is that it provides a snapshot AUC for the sampling period, which is beneficial in groups of patients with high variability in drug pharmacokinetics [2].

First-order pharmacokinetic equations for vancomycin TDM at steady state have been validated [9,10] and are recommended in guidelines [2]. However, the use of this method for vancomycin TDM after the initial dose has not been studied. The aim of this study was to evaluate the usefulness of first-order pharmacokinetic equations for predicting the first-dose vancomycin AUC, and to identify the optimal sampling times to be used in the calculation. We hypothesized that an equation using two appropriate concentration time points could approximate the AUC of the initial vancomycin dose calculated by intensive sampling with reasonable precision, low bias, and high correlation.

## 2. Results

We used two previously published full pharmacokinetic data sets to evaluate the pharmacokinetics of initial dose vancomycin. These were an adult data set from 10 septic shock patients [11], and a pediatric data set from 14 children with severe infection [12]. In the adult data set, the median age was 59 (IQR 45.8–79.5) years and median creatinine clearance was 43.5 (IQR 28.8–92.5) mL/min. An initial vancomycin dose of 30 mg/kg was infused over 120 min. In the pediatric data set, the median age was 6.4 (IQR 3.3–10.7) years and the median creatinine clearance was 183.1 (IQR 148.2–219.5) mL/min/1.73 m^2^; the initial vancomycin dose of 15 mg/kg was infused over 60 min. Detailed characteristics of these studies are described in Appendix A. Details about the building of pharmacokinetic models for the reference standard and two first-order pharmacokinetic equations are provided in the Materials and Methods section.

The analysis results from the adult data set using Model 1 are shown in Appendix A. In this analysis, there were no equation-paired concentration time points with an acceptable mean difference from the Bland–Altman analysis. In contrast, applying Model 2 to the adult data set (Appendix A), C_1_ sampled at 0, 40, 60, and 90 min after the completed infusion combined with C_2_ at 240 min after the completed infusion gave an acceptable −4.6 to 4.1 percent mean difference. The correlation and agreement were better using C_1_ obtained 60 and 90 min after the end of the infusion (Pearson’s r = 0.976 and Lin’s correlation coefficient = 0.971 and 0.967, respectively) than those obtained at 0 and 40 min (Pearson’s r = 0.899, 0.929 and Lin’s correlation coefficient = 0.869 and 0.923, respectively).

In the pediatric data set, Model 1 (Appendix A) using C_2_ at 240 min after the completed infusion, only C_1_ sampled immediately after completing the infusion showed an acceptable −2.7 percent mean difference, with Pearson’s r = 0.974 and Lin’s correlation coefficient = 0.940. Model 1 using C_2_ sampled 300 min after the completed infusion and C_1_ at 0 and 15 min gave 3.4 and −4.8 percent mean difference with Pearson’s r = 0.977, 0.98 and Lin’s correlation coefficient = 0.965, 0.960, respectively.

Appendix A shows the results from the pediatric data set using Model 2. C_1_ between samples 60 and 180 min after the infusion ended resulted in a −4.2 to 2.8 percent mean difference when C_2_ was taken either at 240 or 300 min after the completed infusion. Alternatively, C_1_ sampled 30 min after the completed infusion gave an acceptable percent mean difference only with C_2_ at 240 min after the infusion. In terms of correlation and agreement, all mentioned paired concentration time points had Pearson’s r > 0.96 and Lin’s correlation coefficient >0.95, with the exception of C_1_ at 180 min paired with C_2_ at 240 min after the completed infusion and C_1_ at 180 min paired with C_2_ at 300 min after the completed infusion, both of which had lower correlation and agreement (Pearson’s r = 0.883 and Lin’s correlation coefficient = 0.881; Pearson’s r = 0.939 and Lin’s correlation coefficient = 0.933, respectively).

Selected equation-paired concentration time points with an acceptable mean difference <5% from the Bland–Altman analysis from both the adult and pediatric data sets are shown in Table 1.

Considering all results and looking for intersected equation-paired concentration time points that provided an acceptable percent mean difference, Model 2 consistently provided good agreement, low bias, and high correlation at C_1_ sampled in a time window from 60 to 90 min after the completed infusion, and C_2_ sampled in a time window from 240 to 300 min after the completed infusion (Figure 1). Bland–Altman plots for the selected equation-paired concentration time points are shown in Figure 2.

The pooled data available from Model 2 with C_1_ sampled from 60 to 90 min, and C_2_ from 240 to 300 min after the completed infusion, showed a strong correlation and goodness of fit between the reference standard and the proposed selected equation-paired concentration time point combinations. Figure 3 shows the regression line of best fit between the two methods. The regression line had a slope of 1.03 and an intercept of 0.15, with a correlation coefficient between the two methods of r = 0.994 (*p* < 0.001).

Abbreviations: AUC_f_, full area under the plasma concentration-time curve.

In addition to the full area under the plasma concentration time curve (AUC_f_) calculation, the first-order pharmacokinetic equations were also used for calculating other pharmacokinetic parameters including vancomycin clearance (Cl), volume of distribution (Vd), and half-life (t_1/2_). The corresponding mean and standard deviation for these parameters, as well as the reference standards, are shown in Table 2 and Table 3.

## 3. Discussion

We used two previously published intensive pharmacokinetic data sets and two first-order pharmacokinetic equations to assess which paired post-infusion concentration time point combinations could best predict the actual vancomycin AUC_f_. We found that Model 2 with a compensation area for the alpha-phase, vancomycin serum concentrations collected from 60 to 90 min after the completed infusion as C_1_, and those collected 240–300 min after the completed infusion as C_2_ produced a calculated AUC_f_ with less than 5% mean difference compared to the full linear-log trapezoid method. Pearson’s correlation coefficient and Lin’s agreement coefficient for these combinations were >0.96. The mean difference and 95% limit of agreement for this equation-paired concentration time points combination were sufficiently narrow for clinical purposes. These findings were consistent with whether the infusion period was one or two hours, and regardless of the difference in kidney function in the two data sets. As shown in Table 2 and Table 3, other pharmacokinetic parameters, including vancomycin Cl, Vd, and t_1/2_, showed comparable results with the log-linear trapezoid method. These findings suggested that Model 2 is highly relevant for determining pharmacokinetic parameters after the first dose of vancomycin.

We focused on using two time points within 6 h after the start of the infusion for two reasons. First, this would allow for early dose adjustment, if necessary, in critically ill patients who need to reach the target AUC as soon as possible because achieving the appropriate target of AUC values can reduce mortality in critically ill patients [13]. Second, using two time points within the first six hours makes this approach applicable to both adult and pediatric patients, since the dosing interval in pediatric patients is usually shorter at 6–8 h.

A previous study found that first-order pharmacokinetic equations accurately predicted aminoglycoside AUC [14], daptomycin AUC [15], and steady state vancomycin AUC [9]. The consistency of our results with these other antimicrobial agents is likely due to the similarity in concentration time profile shape.

Pai and Rodvold reported the AUC for aminoglycosides calculation using a first order-pharmacokinetic equation analogous to Model 1 in our study, which had a percent bias of −2.80 (95% CI = −3.06 to −2.54) [14]. In addition, they found that the same method could estimate the AUC of daptomycin with a bias of no more than ±10% [15]. These levels of bias were not expected to be clinically significant; as a result, these methods are widely used for optimizing the administration of aminoglycoside and daptomycin in clinical practice.

Pai et al. also conducted an approach using first-order pharmacokinetic equations to calculate the steady-state vancomycin AUC [9]. The authors proposed two equations analogous to those of Model 1 and Model 2 in our study. In steady state both models showed good agreement with median error < 2%, but their Model 1 tended to underestimate the prediction of the AUC versus the reference standard, consistent with the results of our study.

The most likely reason that Model 1 underestimates AUC versus the reference standard is because this model fails to capture the alpha-phase of the concentration-time curve (Appendix A). Vancomycin is infused over one or two hours, and this equation uses concentration time values that occur after the infusion period; therefore, data from the alpha distribution phase are excluded, and this has a significant impact on the AUC_f_ calculation. As shown in Appendix A, Model 2 captures additional area that at least partially compensates for the alpha-phase AUC, which is not captured by Model 1, resulting in lower bias and improved prediction.

The main advantages of using first-order pharmacokinetic equations are their simplicity and generalizability. Calculations are easy to make using basic computer calculation software, so the method can be easily applied in routine clinical practice. The first-order pharmacokinetic equation method relies primarily on accurate sampling times and concentration measurements, not on the patient’s clinical status, as the absolute rate of vancomycin elimination is a linear function, as previously mentioned [8]. Snapshot measurements with minimal assumptions are especially useful in patients with dynamic clinical states, for example septic patients. As demonstrated in this study, equation-paired concentration time point combinations could accurately predict pharmacokinetic parameters in both intensive pharmacokinetic data sets, despite the disparate demographic characteristics, kidney function, initial infusion dose, and infusion period.

There are a number of limitations to our research. First, although the target AUC for the first dose of vancomycin is unclear, DeRyke and Alexander [16] demonstrate that pharmacokinetic parameters derived from pharmacokinetic equations can be used to derive an area under the plasma concentration-time curve in the first 24 h of treatment (AUC_0–24_) for each individual patient. Calculating pharmacokinetic parameters based on the first vancomycin dose can be used for optimizing the subsequent doses using the method described by Sawchuk-Zaske [17]. Several studies using this method to individualize vancomycin regimens showed that first-dose TDM resulted in faster target AUC attainment than routine steady-state TDM in various group of patients, including critically ill adults and neonates [18,19,20]. Second, our study aimed to identify the best-performing first-order pharmacokinetic equation and optimal sampling times that reliably reproducibly estimated first dose vancomycin AUC and other pharmacokinetic parameters using two independent data sets. The data sets we used to compare the actual AUC against the AUC predicted by each equation were from a group of adults with septic shock and a group of children with severe infection (Appendix A). Our findings may therefore need further evaluation before generalization to other groups of patients, such as those who are overweight/obese, and those with renal impairment. Lastly, the data sets used in our study were comparatively small, and resulted in wide standard deviations around the mean difference in Bland-Altman analysis. Although conducting a new similar study with a larger sample size would likely lower this variability, accessing intensive pharmacokinetic data sets to make these calculations is difficult and designing a new study is an ethically challenging because the subjects are vulnerable. Furthermore, the 95% limit of agreements of selected equation-paired concentration time point combinations in our study was not expected to be clinically significant. Therefore, despite the limitations mentioned and the small sample size, our current study results provide information which can guide clinical practice in some groups of patients, and pharmacokinetic parameters based on the first vancomycin dose are useful for further AUC_0–24_ research study (TCTR20210617001).

## 4. Materials and Methods

The study flow diagram of this study was presented in Figure 4.

### 4.1. Pharmacokinetic Model Building

#### 4.1.1. The Reference Standard

The reference AUC_f_ of the individuals in both studies were calculated using all plasma concentrations available (10 time points for subjects in the adult data set and 9 time points for subjects in the pediatric data set, Appendix A) and the linear-log trapezoid rule. The pharmacokinetic analysis of the serum concentration-time data was performed using PKSolver (version 2.0, China Pharmaceutical University, Nanjing, China) [21].

#### 4.1.2. The First-Order Pharmacokinetic Equations [9]

Using two available serum vancomycin concentrations, the elimination rate constant (Ke) can be calculated using the following equation [22]:(1)Ke=LnC1C2t,
where C_1_ is the first concentration measured after the infusion has been completed, C_2_ is the second concentration collected toward the end of the dosing interval, and t is the difference in time between C_1_ and C_2_. Once the Ke is computed, it can be used to compute theoretical concentrations through forward- and backward-extrapolation.

In the adult data set, vancomycin was infused over 120 min and concentration levels were sampled at 0, 10, 20, 40, 60, 90, 120, and 240 min after the completed infusion. C_1_ was sequentially chosen from 0 to 120 min after the completed infusion and C_2_ was the last sample, collected 240 min after the completed infusion. In the pediatric data set, vancomycin was infused over 60 min and levels were sampled at 0, 15, 30, 60, 120, 180, 240, and 300 min after the completed infusion. C_1_ was sequentially chosen from 0 to 240 min after the completed infusion and C_2_ was chosen from 240 or 300 min after the completed infusion.

In this study, we proposed and compared two first-order pharmacokinetic equation models analogous to those used by Pai et al. in their study on optimizing vancomycin delivery at steady-state [9]. The models can be described as follows:Model 1 (Appendix A)

For the first dose, the vancomycin concentration at the start of infusion is zero. The end of infusion concentration is estimated (C_eoi′_) regardless effect of the alpha-phase of the 2-compartment distribution model. The area between the start and the end of the infusion time (t_eoi_) can be measured as the area of the triangle:(2)$${\mathrm{AUC}}_{0-t_{\mathrm{eoi}}}=\frac{t_{\mathrm{eoi}}\times C_{\mathrm{eoi}\prime }}{2}$$

The area under the exponential part from t_eoi_ to the end of dose (t_∞_) is:(3)$${\mathrm{AUC}}_{t_{\mathrm{eoi}}-t_{\infty }}=\frac{C_{\mathrm{eoi}\prime }}{\mathrm{Ke}}$$

Therefore, the AUC_f_ for first dose vancomycin in Model 1 can be simplified to:(4)$${\mathrm{AUC}}_{f}=\frac{t_{\mathrm{eoi}}\times C_{\mathrm{eoi}\prime }}{2}+\frac{C_{\mathrm{eoi}\prime }}{\mathrm{Ke}}$$


Model 2 (Appendix A)


This model captures additional area by backward extrapolating peak concentration to the start of infusion (C_0′_), aiming to compensate for the unmeasured alpha-phase. Under this model, the equation can be simplified to:(5)$${\mathrm{AUC}}_{f}=\frac{C_{0\prime }}{\mathrm{Ke}}$$

First-order pharmacokinetic equation calculations were performed using Microsoft Office Excel 365 (Redmond, Washington, DC, USA).

### 4.2. Statistical Analysis

To evaluate Equations (4) and (5), the linear-log trapezoid rule was used to calculate the AUC_f_ using all available samples from the full pharmacokinetic studies, and served as the reference standard. Bland-Altman analysis was used to assess agreement and bias [23] between the reference standard and each combination of the first-order pharmacokinetic equations and selected paired concentration time points. To facilitate the comparison of the equation-derived and gold standard AUC from the adult and pediatric studies, results were expressed as percent change. An a priori acceptable mean difference was set at 5%. Pearson’s correlation and linear regression were used to assess the linear correlation of estimates. Lin’s correlation coefficient was also used to assess agreement. Statistical significance was defined as *p* < 0.05. Furthermore, a summary of pharmacokinetic parameters derived from selected equation-paired concentration time points combination including vancomycin Cl, Vd, and t_1/2_ were calculated and compared with the parameters derived from the linear-log trapezoid rule, which served as the reference standard. All statistical analyses were performed using Stata version 15.1 (Stata Corp LCC, College Station, TX, USA).

## 5. Conclusions

Of the two first-order pharmacokinetic equations tested, the one that used plasma samples obtained from 60 to 90 and 240 to 300 min after the completed infusion reliably and reproducibly estimated first dose vancomycin AUC. If physicians used this equation in clinical practice settings, doses could be adjusted earlier than in conventional practice and lead to a more rapid achievement of the target AUC. However, whether achieving the target AUC earlier would definitively improve patient outcomes by enhancing the therapeutic effect and reducing adverse drug effects needs further investigation.

## Figures and Tables

**Figure 1 antibiotics-12-00630-f001:**
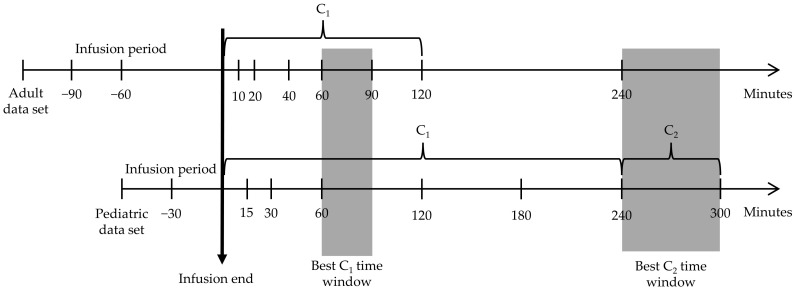
Summary of intersected C1 and C2 windows as inputs of the first-order pharmacokinetic equation using Model 2.

**Figure 2 antibiotics-12-00630-f002:**
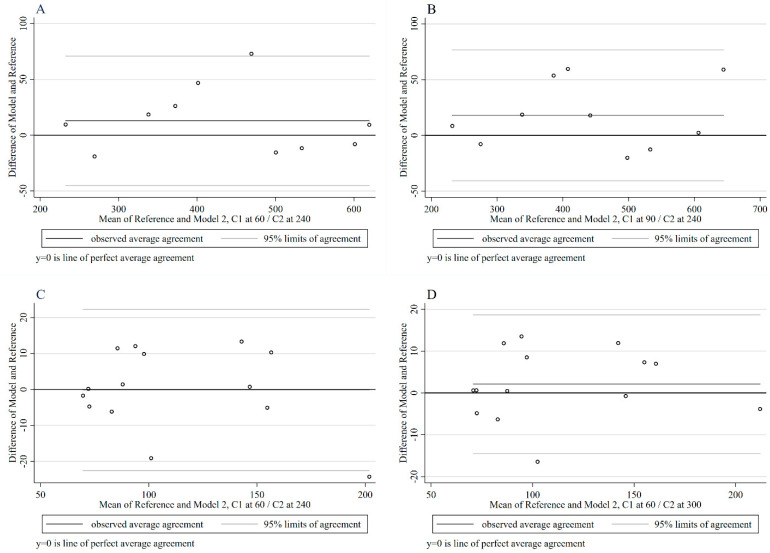
Bland-Altman plots of the reference standard and each selected equation-paired concentration time points combination. (**A**) Adult data set using Model 2 with C1 at 60 min and C2 at 240 after the completed infusion had mean bias of 13.0 (3.0%) and 95% limits of agreement of −45.0 to 70.9. (**B**) Adult data set using Model 2 with C1 at 90 min and C2 at 240 after the completed infusion had mean bias of 18.0 (4.1%) and 95% limits of agreement of −40.8 to 76.7. (**C**) Pediatric data set using Model 2 with C1 at 60 min and C2 at 240 after the completed infusion had bias of −0.1 (−0.1%) and 95% limits of agreement of −22.6 to 22.3. (**D**) Pediatric data set using Model 2 with C1 at 60 min and C2 at 300 after the completed infusion had bias of 2.1 (1.9%) and 95% limits of agreement of −14.4 to 18.7.

**Figure 3 antibiotics-12-00630-f003:**
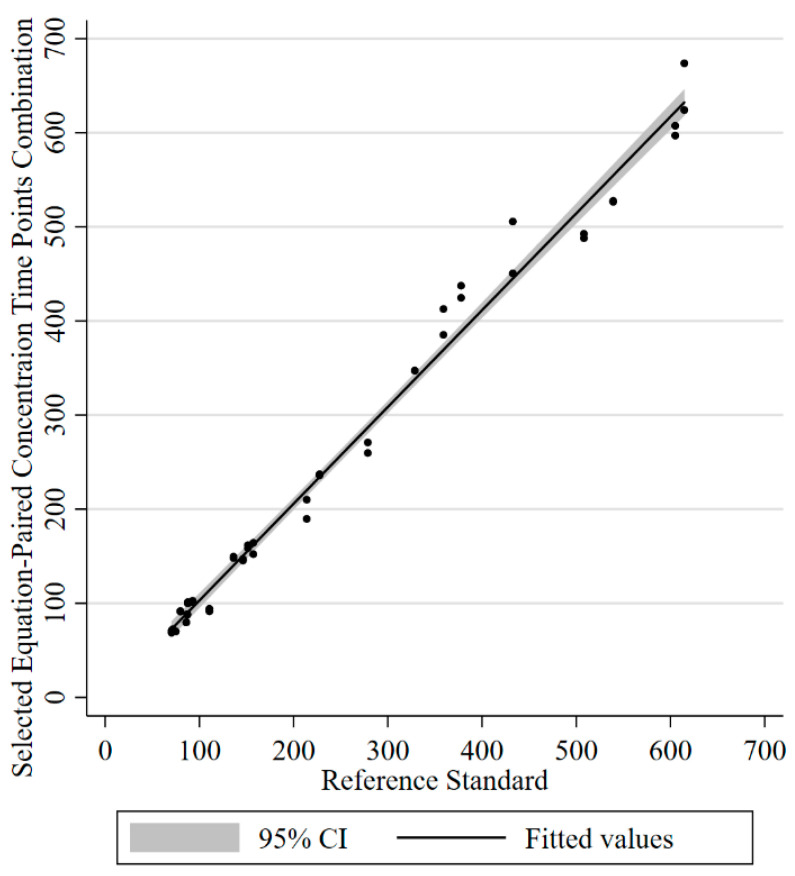
The regression line between the reference standard AUC_f_ and pooled selected equation-paired concentration time point combinations. The regression equation was y = 0.15 + 1.03x (R^2^ = 0.988, *p* < 0.001). Pearson’s correlation coefficient between the two methods was r = 0.994 (*p* < 0.001).

**Figure 4 antibiotics-12-00630-f004:**
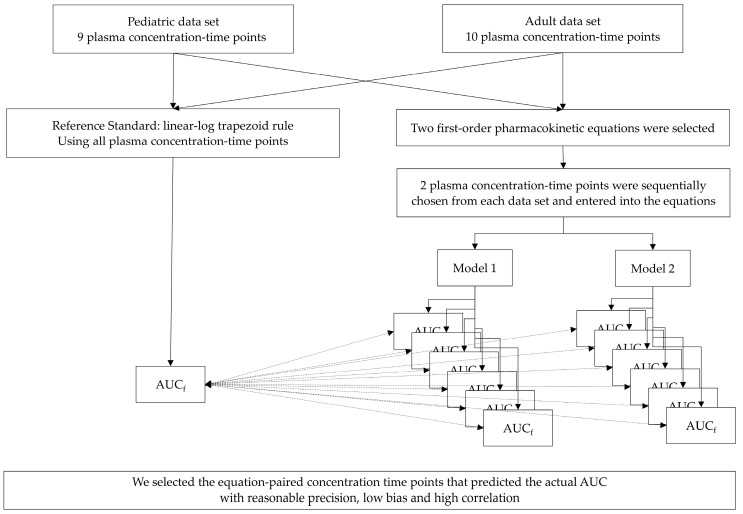
Study flow diagram. Abbreviations: AUC_f_, full area under the plasma concentration-time curve.

**Table 1 antibiotics-12-00630-t001:** Selected equation-paired concentration time points with an acceptable mean difference from the Bland–Altman analysis from adult and pediatric data sets.

Data Set	Time Points		Bland–Altman Analysis				Correlation		Lin’s Coefficients	
	C_1_	C_2_	Mean	Difference		95% Limits of Agreement	Pearson’s r	*p*-Value	Rho_c	*p*-Value
				Mean (%)	SD					
Model 1										
Pediatric	0	240	111.9	−3.0 (−2.7)	13.4	(−29.1, 23.2)	0.974	<0.001	0.940	<0.001
	0	300	115.3	3.9 (3.4)	10.2	(−16.2, 24.0)	0.977	<0.001	0.965	<0.001
	15	300	109.3	−5.3 (−4.8)	9.9	(−24.7, 14.2)	0.98	<0.001	0.960	<0.001
Model 2										
Adult	0	240	417.5	−19.3 (−4.6)	60.2	(−137.3, 98.8)	0.899	<0.001	0.869	<0.001
	40	240	422.2	−9.9 (−2.3)	50.0	(−107.9, 88.0)	0.929	<0.001	0.923	<0.001
	60	240	433.6	13.0 (3.0)	30.0	(−45.0, 70.9)	0.976	<0.001	0.971	<0.001
	90	240	436.1	18.0 (4.1)	30.0	(−40.8, 76.7)	0.976	<0.001	0.967	<0.001
Pediatric	30	240	114.1	4.3 (3.8)	11.3	(−17.8, 26.4)	0.969	<0.001	0.955	<0.001
	60	240	111.8	−0.1 (−0.1)	11.4	(−22.6, 22.3)	0.964	<0.001	0.962	<0.001
	120	240	109.6	−4.6 (−4.2)	11.6	(−27.4, 18.2)	0.963	<0.001	0.957	<0.001
	180	240	113.3	2.8 (2.5)	20.5	(−37.4, 43.0)	0.883	<0.001	0.881	<0.001
	60	300	113.0	2.1 (1.9)	8.4	(−14.4, 18.7)	0.981	<0.001	0.979	<0.001
	120	300	109.7	−4.4 (−4.01)	10.6	(−25.1, 16.2)	0.974	<0.001	0.966	<0.001
	180	300	113.5	3.2 (2.8)	16.2	(−28.5, 34.9)	0.939	<0.001	0.933	<0.001

**Table 2 antibiotics-12-00630-t002:** Summary of pharmacokinetic parameters of the adult data set derived from selected equation-paired concentration time point combinations compared to the reference standard.

Adult Data Set	Reference Standard	Model 2	
		C_1_ at 60 and C_2_ at 240 min after the Completed Infusion	C_1_ at 90 and C_2_ at 240 min after the Completed Infusion
AUC, mean ± SD (mg/L × h)	427.14 ± 135.26	440.11 ± 132.85	445.09 ± 138.21
Vancomycin Cl, mean ± SD (L/h)	4.62 ± 1.45	4.47 ± 1.38	4.43 ± 1.40
Vd, mean ± SD (L)	39.35 ± 8.95	39.88 ± 8.92	41.33 ± 7.76
Half-life ± SD (h)	6.28 ± 1.97	6.56 ± 2.02	6.83 ± 1.69

Abbreviations: AUC, area under the concentration curve; Cl, clearance; Vd, volume of distribution.

**Table 3 antibiotics-12-00630-t003:** Summary of pharmacokinetic parameters of the pediatric data set derived from selected equation-paired concentration time point combinations compared to the reference standard.

Pediatric Data Set	Reference Standard	Model 2	
		C_1_ at 60 and C_2_ at 240 min after the Completed Infusion	C_1_ at 60 and C_2_ at 300 min after the Completed Infusion
AUC, mean ± SD (mg/L × h)	111.91 ± 42.73	111.78 ± 39.83	114.03 ± 43.39
Vancomycin Cl, mean ± SD (L/kg/h)	0.16 ± 0.04	0.16 ± 0.04	0.15 ± 0.04
Vd, mean ± SD (L/kg)	0.55 ± 0.10	0.51 ± 0.06	0.52 ± 0.07
Half-life ± SD (h)	2.58 ± 0.91	2.38 ± 0.69	2.47 ± 0.84

Abbreviations: AUC, area under the concentration curve; Cl, clearance; Vd, volume of distribution.

## Data Availability

No new data were created in this study. Data sharing is not applicable to this article.

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
