# Peer review of "Predicting the Area under the Plasma Concentration-Time Curve (AUC) for First Dose Vancomycin Using First-Order Pharmacokinetic Equations"

_antibiotics, 2023, doi:10.3390/antibiotics12040630_

Round 1

Reviewer 1 Report

This paper described that the simplified method for calculating area under the plasma concentration-time curve (AUC) for first dose vancomycin using a first vancomycin serum concentration from 60-90 minutes and the second concentration from 240-300 minutes after the completed infusion.

The obtained results may be useful for the management of vancomycin serum concentrations, however, their usefulness appears to be limited by practicality of the equation.

1) It is certain that the AUC for first dose vancomycin is calculated by the first-order pharmacokinetic equation models proposed in this study. However, the target AUC at the first dose vancomycin is unclear. It is also not possible to estimate the AUC or optimal dosage for the second and subsequent doses of vancomycin. The authors have to clearly demonstrate how the calculation of the AUC for first dose vancomycin using the proposed equation is clinically useful.

2) Can the proposed equation be used for all adult and pediatric patients, including obese and critically ill patients? The authors should correctly indicate the limitations not only of the sample size but also of the equation based on the data sets.

Author Response

Point 1: It is certain that the AUC for first dose vancomycin is calculated by the first-order pharmacokinetic equation models proposed in this study. However, the target AUC at the first dose vancomycin is unclear. It is also not possible to estimate the AUC or optimal dosage for the second and subsequent doses of vancomycin. The authors have to clearly demonstrate how the calculation of the AUC for first dose vancomycin using the proposed equation is clinically useful.

Response 1: Thank you for these comments.  We have re-written sections of our paper to address these points.  Our study demonstrated that selected equation-paired concentration time point combinations can used for calculating the AUC for the first dose vancomycin as well as other pharmacokinetic parameters including vancomycin clearance (Cl), volume of distribution (Vd), and half-life (t1/2). Although the target AUC at the first dose vancomycin is unclear, DeRyke and Alexander methods [1] demonstrate that pharmacokinetic parameters derived from pharmacokinetic equations can be used to derive an area under the plasma concentration-time curve in the first 24 hours of treatment (AUC0-24) for each individual patient. Moreover, calculating pharmacokinetic parameters based on the first vancomycin dose can be used for optimizing the subsequent doses using the method described by Sawchuk-Zaske [2]. Several studies using this method to individualize vancomycin regimens showed that first-dose therapeutic drug monitoring resulted in faster target AUC attainment than routine steady-state therapeutic drug monitoring in various group of patients including critically ill adults and neonates [3-5]. Our study aimed to identify the best-performing first-order pharmacokinetic equation and optimal sampling times that reliably and reproducibly estimated first dose vancomycin AUC and these useful pharmacokinetic parameters using on two independent data sets.  We have modified our discussion to clarify these points. 

Point 2: Can the proposed equation be used for all adult and pediatric patients, including obese and critically ill patients? The authors should correctly indicate the limitations not only of the sample size but also of the equation based on the data sets.

Response 2: Although the first-order pharmacokinetic equation provided individual level information for each subject [6], the data sets we used included only adults with septic shock and pediatric patients with severe infection; the demographic characteristics of these patients are described in table S1.  Generalization of our results to other populations needs further evaluation. We have expanded the limitations section of the paper to note these important ideas noted by the reviewer in points 1 and 2.

References

  1. Deryke, C. Andrew, and Donald P. Alexander. "Optimizing Vancomycin Dosing through Pharmacodynamic Assessment Targeting Area under the Concentration-Time Curve/Minimum Inhibitory Concentration." Hospital Pharmacy 44, no. 9 (2009): 751-65.
  2. Sawchuk, R. J., D. E. Zaske, R. J. Cipolle, W. A. Wargin, and R. G. Strate. "Kinetic Model for Gentamicin Dosing with the Use of Individual Patient Parameters." Clin Pharmacol Ther 21, no. 3 (1977): 362-9.
  3. Shahrami, B., F. Najmeddin, S. Mousavi, A. Ahmadi, M. R. Rouini, K. Sadeghi, and M. Mojtahedzadeh. "Achievement of Vancomycin Therapeutic Goals in Critically Ill Patients: Early Individualization May Be Beneficial." Crit Care Res Pract 2016 (2016): 1245815.
  4. Truong, J., S. R. Smith, J. J. Veillette, and S. C. Forland. "Individualized Pharmacokinetic Dosing of Vancomycin Reduces Time to Therapeutic Trough Concentrations in Critically Ill Patients." J Clin Pharmacol 58, no. 9 (2018): 1123-30.
  5. Crumby, T., E. Rinehart, M. C. Carby, D. Kuhl, and A. J. Talati. "Pharmacokinetic Comparison of Nomogram-Based and Individualized Vancomycin Regimens in Neonates." Am J Health Syst Pharm 66, no. 2 (2009): 149-53.
  6. Pai, M. P., M. Neely, K. A. Rodvold, and T. P. Lodise. "Innovative Approaches to Optimizing the Delivery of Vancomycin in Individual Patients." Adv Drug Deliv Rev 77 (2014): 50-7.

Reviewer 2 Report

Title of the paper is too ambiguous to understand what the authors are trying to convey.

In the last paragraph of the introduction section, it is recommended that the rationale of the research must be incorporated in the form of the figure.

Conclusion must be improved

Round 2

Reviewer 1 Report

The manuscript was revised according to the reviewers’ comments. However, since this paper has many supplementary data, it is better to reconsider those.

Author Response

Thank you for the comment. We have moved figure S1 from supplementary materials to the main text. For other supplementary data, we think they are necessary and justified for the sake of transparency.

Reviewer 2 Report

All the comments were addressed by the authors and thus improved the level of manuscript. Additionally, it is recomended that the rationale figure must be incorportaed in main text not in supplementary data.

Author Response

Thank you for the comment. We have moved figure S1 from supplementary materials to the main text as figure 4.